# Biochemical response of *Sonneratia alba* Sm. branches infested by a wood boring moth (Gazi Bay, Kenya)

**Elisha Mrabu Jenoh**[1,2]*, **Mohamed Traoré**[3], **Charles Kosore**[1], **Nico Koedam**[2]

**1** Kenya Marine and Fisheries Research Institute (KMFRI), Mombasa, Kenya, **2** Laboratory of Plant Biology and Nature Management (APNA), Ecology & Biodiversity, Vrije Universiteit Brussel, Brussels, Belgium, **3** Department of Geology and Mines, Ecole Nationale d'Ingénieur–Abderhamane Baba Touré, Bamako, Mali

* elishamrabu@gmail.com

**Data Availability Statement:** All relevant data are within the manuscript and its Supporting Information files.

**Funding:** This research was funded by the Flemish Interuniversity Council—University Development

## Abstract

Infestation by a moth woodborer species is causing mortality of *Sonneratia alba* Sm. mangrove by tunneling through the inner bark, cambium and conductive tissue. Infestation leads to death of some infested branches, whereas in other cases infested branches have been observed to recover from infestation. We have used Fourier transform infrared spectroscopy (FTIR) to investigate the differences in macromolecule (polysaccharide and lignin) content present in branches that died (D) of the infestation, those that recovered (R) from the infestation and control branches (C) that were not subject to any infestation. Wood samples were taken from four sampling plots (A, B, C and D) in Gazi Bay (Kenya). From each of the four plots, 15 *S. alba* branches were taken from five trees, from which 1 cm thick discs were cut from each of these branches to be used as samples. To identify the most characteristic FTIR bands for the three groups of samples, Principal Component Analysis (PCA) was applied on the transposed data matrix. Furthermore, canonical discriminant analysis was applied on the data considering the main FTIR band that would be identified through the PCA factors. Finally, One-way ANOVA and post hoc test were used to verify the significance of the observed trends. Branches that recovered from infestation had higher relative abundance of lignified cells. We conclude that insect-infested *S. alba* undergo changes related to the lignocellulosic contents. The infestation induces a decrease of the proportion of the polysaccharide content and an increase of the proportion of the lignin contents.

## Introduction

The infestation by a lepidopteran moth woodborer species is causing mortality of *Sonneratia alba* Sm., which is an ubiquitous and pioneer mangrove specie in the family Lythraceae. This mangrove tree occurs along the waterfront in a variety of mangrove ecosystem settings of the Indo-Pacific region [1, 2]. An unknown wood boring larvae of a moth has been found to be responsible for the infestation of this species in Gazi Bay (Kenya). However, ongoing work of scrutinizing and typification has placed the moth to an undescribed genus named "Gen. Nov.

Cooperation (VLIR-UOS; http://www.vliruos.be/) ICP Ph. D Scholarship 2011. The funder had no role in study design, data collection and analysis, decision to publish, or preparation of the manuscript.

**Competing interests:** The authors have declared that no competing interests exist.

ZA" that represents its type species named "sp. nov. za" [3]. The description of this new genus with its detailed presentation and names of all currently known species from the Afrotropical Region is currently being dealt with in a publication by Lehmann, Jenoh, Kioko and Koedam (in prep.). The insect infests the mangrove by tunnelling through the inner bark, cambium and conductive tissue; if the branch is completely girdled, the branch dies at above the damaged site [1, 2, 4]. Indeed, many species of Lepidoptera spend most of their larval development tunnelling and feeding in the stems and branches of trees and shrubs [2, 4, 5]. Partial girdling reduces tree growth and vigour above the site of attack. Infestation can lead to the death of the infested branches and in some severe cases where other factors (e.g. nutrient limitation, salinity increase, and associated stressors) enhance the damage, thus entire trees die [2, 6]. In some cases, infested *S. alba* branches have been observed to recover from infestation whereas other branches are unable to survive the infestation [2, 4].

Mangrove ecosystems are of great ecological and economic importance in the Western Indian Ocean (WIO) region, degradation of *S. alba* may lead to the loss of diversity [7]. And also its degradation could contribute to an increase in coastal erosion since it is the waterfront pioneer species, hence it assists the other more socio-economically important species to establish by buffering the mangrove formation from the open sea [8]. Furthermore, since the mangrove at the margin of water have been found to have high carbon sequestration ability, the infestation of *S. alba* is a danger to the entire mangrove forest and may lead to highly reduce its ability of carbon sequestration [9, 10]. Owing to these vital ecological roles, it is paramount that information to enhance conservation of this species against insect infestation be availed for a sustainable management of the entire forest and eventual providence of quality goods and services of the mangrove to the coastal community.

After an infestation, the dying or recovery process of a branch is dependent on factors that are related to the strength and the timeliness of the primary and secondary plant defences against the infesting agent. Primary defences include structural traits such as an intact and impenetrable barrier composed of bark and a waxy cuticle, trichomes, and cell wall thickness and lignification [11, 12]. Lignin is a natural aromatic polymer, it enhances plant cell wall rigidity, hydrophobic properties and promotes minerals transport through the vascular bundles in plants. Also, lignin compounds are indicated as important barrier that protects plants against pests and pathogens [13, 14]. During infestation, plants increase deposition of lignin as a response to a variety of biotic and abiotic stresses, even though lignin deposition in plant cell walls is also a normal developmental process [13, 15, 16]. The success of an infesting woodborer is therefore pegged on its ability to overcome the rigidity of deposited lignin and other defence chemicals in the plant wood structure.

Fourier transform infrared (FTIR) spectroscopy is a useful analytical technique for wood structural chemistry characterization with just a minimum sample preparation required. This technique has been widely used in studies on characterization of lignocellulosic materials [17–21]. The application of FTIR on wood samples allow to get details on functional groups, molecular bonds and specific structural features of wood chemical contents [21–24]. Also, the use of the FTIR permitted to identify differences in wood parts (sapwood and heartwood), wood type (softwood and hardwood) and also assessment of wood quality [25–27]. FTIR has been successfully used to assess variations in the relative proportions of carbohydrates and lignin in eucalyptus wood [28, 29] and to identify different tree species such as hardwood and softwood by considering the relative proportions of polysaccharides and lignin compounds [23, 29, 30].

The main aim of this work is to get understanding about the influence of moth infestation on *S. alba* wood lignocellulosic contents. In this regard, we have used FTIR to investigate the differences in chemical contents present in three different groups of *S. alba* wood samples. The first group (D) consists of branches that died from the moth infestation, the second group (R)

represents branches that were infested by the moth and recovered from the infestation, and the third group (C) refers to the control branches that were not subject of any infestation by the moth. Then we discussed the determinant role of polysaccharide and lignin compounds in the death/recovery processes of the infested branches.

## Material and methods

### Sampling sites and samples

Wood samples were taken from Gazi Bay, located at about 55 km south of Mombasa in Kwale County (Fig 1, Table 1), where all mangrove tree species that occur in Kenya are present [31]. Four sampling plots (A, B, C and D; see Table 1) were chosen by considering the availability of a relatively dense *S. alba* forest. It is important to note that in the study site, and Kenya in general, *S. alba* forms only narrow fringing areas in the mangrove [31, 32]. However, it is often dominant at the mangrove's water edge and on mudflats. Plot A was composed of two sections, the first with large old *S. alba* trees (natural occurring) and the second with *S. alba* forest which was replanted on an originally forested site, thirteen and ten years at the time of sampling, respectively. Plot B was mainly composed of old *S. alba* naturally grown trees. In addition to insect infestation, this plot has been affected by sedimentation and hence lost several trees. Plot C was a 20-year-old *S. alba* plantation. Plot D was composed of a mature, naturally grown *S. alba* forest located alongside the main creek in Gazi Bay.

From each of the four plots, 15 *S. alba* branches were taken from five trees at low tides during the month of February 2019. From each of these trees, dead (D), recovered (R) and control (C) branches were cut. Efforts were made to ensure the branches were of equal size and were situated in the same canopy position. One (1) cm thick discs were then cut from each of these branches to be used as samples (Fig 2). The discs were oven dried at 30˚C for two weeks before being used for FTIR measurement.

### FTIR measurement

Wood spectral measurements were conducted using Fourier transform infrared spectroscopy (FTIR-ATR) analyses, an Agilent Cary 630 FTIR Spectrometer equipped with a single-reflection diamond crystal. The spectra were collected in the absorbance range from 4000 to 400 cm$^{-1}$ over 100 scans per sample, at a resolution of 4 cm$^{-1}$. The angle of incidence for the infrared beam through the diamond crystal was 45˚. For each wood disc, FTIR Spectra were recorded directly on the surface of the wood fragments, at 1 mm interval in consecutive positions from the outer region (near the cambium) to the inner part (toward the pith) of the wood discs.

### Data analysis

Average and standard deviation of the fingerprint region spectra were calculated for general description of the FTIR spectra. This aimed to visualize the wood FTIR data and to observe the most general patterns and changes for these three types of wood samples. The average spectra show the most common vibrations in the samples whereas the standard deviation spectra warn about variations in the samples. To get better insight about the results, Principal Component Analysis (PCA) has been applied on the transposed data matrix (samples as variables, bands as observations). This approach is being common practice in wood analysis, and has been successfully applied to wood molecular studies [26]. And, the extraction of PCA factors was carried out using the varimax rotation. PCA enabled the identification of the most characteristic FTIR bands of the analyzed samples. Thereafter, linear discriminant analysis (LDA)

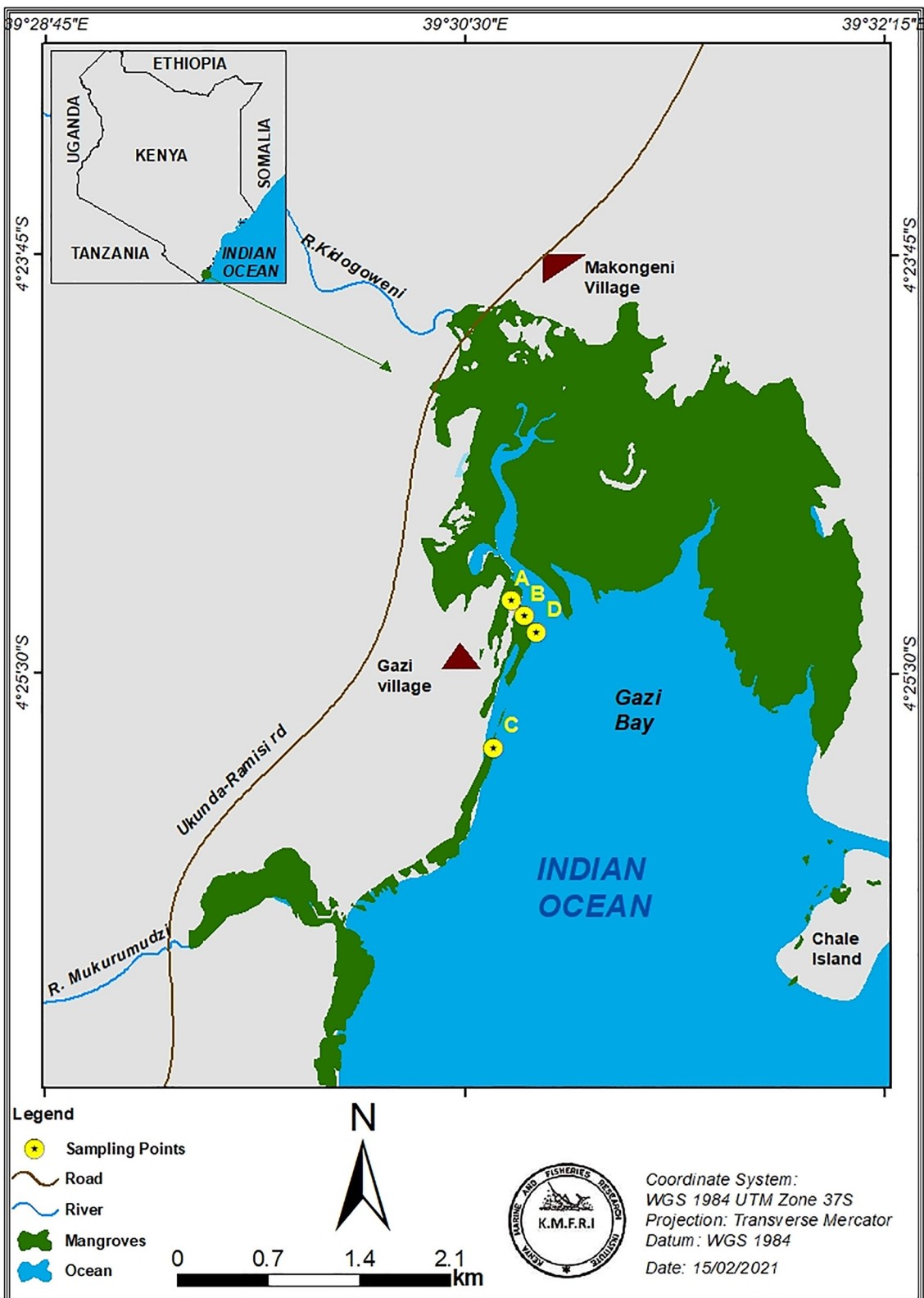

**Fig 1. A map of Kenya (inset) and the sampling sites (in Gazi Bay).** The yellow circles at the sampling sites represent the sampling plots where the study was conducted.

**Table 1. Sampling plots and details of the sampling plots in Gazi Bay.**

| Sampling plots | Geographic location | Comments on the selected samples (as per at the time of sampling) |
|---|---|---|
| A | 04˚25'901"S 039˚30'676"E | Old naturally grown trees and two replanted forest (13 years and 10 years) |
| B | 04˚25'589"S 039˚30'789"E | Old naturally grown trees |
| C | 03˚21'100"S 039˚58'124"E | Replanted forest (20 years old) |
| D | 03˚21'100"S 039˚58'124"E | Mature naturally grown trees |

was applied on the data considering the main FTIR bands identified through the PCA factors. This aimed at supporting the PCA results and also to get better understanding on the molecular characteristics of the studied wood samples. Furthermore, to assess the significance of the differences between samples, the One-way ANOVA test was used; and the use of the post hoc test of Student–Newman–Keuls (with alpha = 0.05) permitted the classification by homogenous subsets. All statistical tests were done using SPSS 23.

## Results

### FTIR data

In Fig 3, the average spectra indicate that all the three samples present similarities according to the absorption bands. Also, at some region (for example between 1580 and 1360 cm$^{-1}$), the values of the relative absorbance appeared very much alike. However, around the regions between 1730 and 1580 cm$^{-1}$ and 1240 and 1180 cm$^{-1}$ the value of the relative absorbance related to the control samples was lower, whereas it was higher around the region between 1115 and 880 cm$^{-1}$. As for the standard deviation spectra, they highlighted the FTIR bands with the highest variability for the samples. Actually, standard deviation spectra for the recovered samples indicated to present the highest variability, as indicated in many of the bands. Separate peaks at bands near 1610, 1191 and 1020 cm$^{-1}$ were well distinguished by the standard deviation spectra, and they were detected within the above mentioned spectral region.

### Principal component analysis

The two first factors of PCA on the transpose data matrix accounted for 99% of the total variance. The score plots for these principal components are shown in Fig 4. PC1 with 54% of the

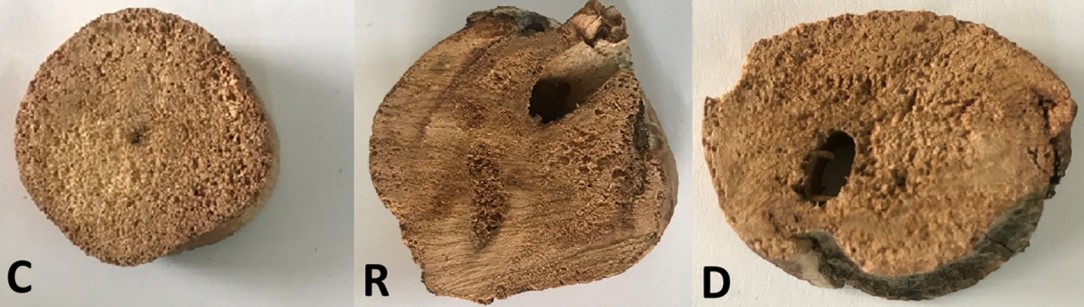

**Fig 2. Pictures of wood discs used for the measurement of spectra.** C stands for a sample from the control branch; R stands for a sample from the recovered branch; and D stands for a sample for dead branch.

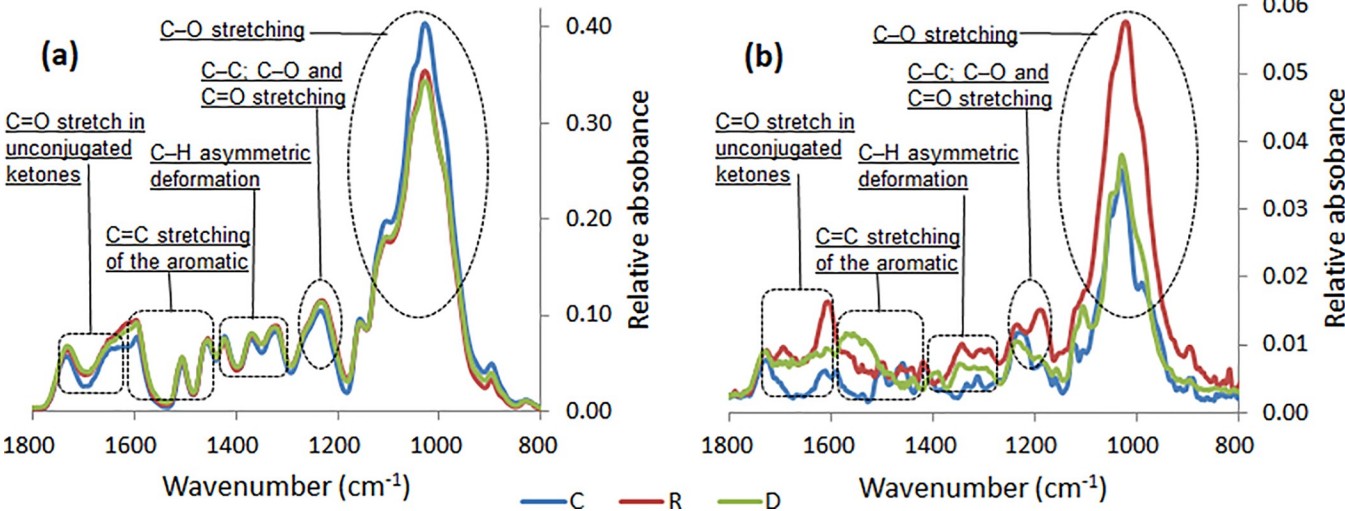

**Fig 3. Average and standard deviation spectra for the fingerprint regions of *Sonneratia alba* wood samples.** (C for control samples; R for recovered samples; and D for died samples).

total variance characterized by the FTIR bands near 1020 and 1053 cm$^{-1}$ with positive scores; and FTIR bands near 1193, 1348, 1445, 1521, 1610 and 1698 cm$^{-1}$ with negative scores. Besides, PC2 with 45% of the total variance was characterized by bands near 1035, 1103, 1220, 1325, 1451 and 1606 cm$^{-1}$ with positive scores; bands near 1408, 1480 and 1542 cm$^{-1}$ with negative scores. All the related FTIR bands and their molecular assignment are provided in Table 2.

The bar-plot of the loadings of the samples for the two first components (Fig 5) shows significant differences ($P < 0.01$; see Table 3) between the control samples and the other two types of sample (recovered and dead wood samples). Control samples were associated to the higher loadings for the first factor (PC1), whereas, they were associated to the lower loadings for the second factor (PC2). Here also the high standard deviation bars indicated variability associated to the recovered samples.

## Discriminant analysis

Discriminant analysis applied on specific FTIR bands associated to the two first PCA factors permitted to separate the three groups of samples (Fig 6). The first discriminant function

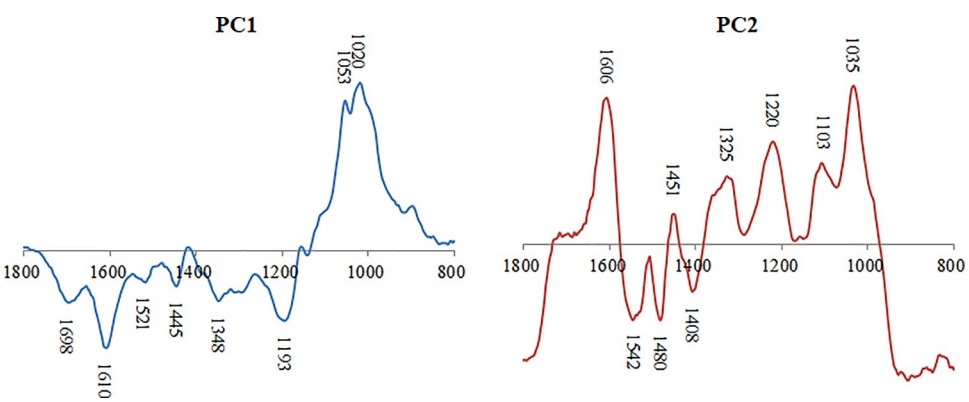

**Fig 4. Component score plot of PC1 and PC2 loadings.**

**Table 2. Infrared bands and related molecular bond assignments according to the literature.**

| Wn (cm⁻¹) | Band assignment | References | PCA factor |
|---|---|---|---|
| 1020 | Polysaccharides | Popescu *et al.*, 2007 | PC1 |
| 1035 | Polysaccharides | Popescu *et al.*, 2007 | PC1 |
| 1053 | Polyssaccharides | Faix, 1991 | PC1 |
| 1103 | Polyssaccharides | McCann *et al.*, 1992; Zhang et al., 2010 | PC2 |
| 1193 | Polysaccharides | Zhou *et al.*, 2015 | PC1 |
| 1220 | Lignin | Chen *et al.*, 2010; Zhou *et al.*, 2015 | PC2 |
| 1325 | Polysaccharides | Colom and Carrillo 2005; Popescu et al. 2007 | PC2 |
| 1348 | Polysaccharides | Evans *et al.*, 1992; Mohebby, 2008 | PC1 |
| 1408 | Polysaccharides | Zhang et al., 2010 | PC2 |
| 1445 | Lignin | Faix, 1991; Zhang *et al.*, 2010 | PC1 |
| 1451 | Lignin | Popescu *et al.*, 2007; Chen *et al.*, 2010 | PC2 |
| 1480 | Lignin | Popescu *et al.*, 2007; Chen *et al.*, 2010 | PC2 |
| 1521 | Polysaccharides | Popescu *et al.*, 2007; Zhou *et al.*, 2015 | PC1 |
| 1542 | Polysaccharides | Popescu *et al.*, 2007; Zhou *et al.*, 2015 | PC2 |
| 1606 | Lignin | Zhao *et al.*, 2014 | PC1 |
| 1610 | Lignin | Zhao *et al.*, 2014 | PC1 |
| 1698 | Polysaccharides | Mizzoni and Cessaro, 2007; Vahur *et al.*, 2011 | PC1 |

(DF1) accounted for about 60% of the total variance and the second discriminant function (DF2) accounted for about 40% of the total variance. According to DF1, the recovered wood samples (with positive scores) were significantly different (*P* < 0.01; again see Table 3) from the control and dead wood samples (with negative scores). And from DF2, all the wood samples appeared to be significantly different (*P* < 0.01; see Table 4). For the latter, the control wood samples were associated to the highest positive scores and the other two types of samples were associated to negative scores.

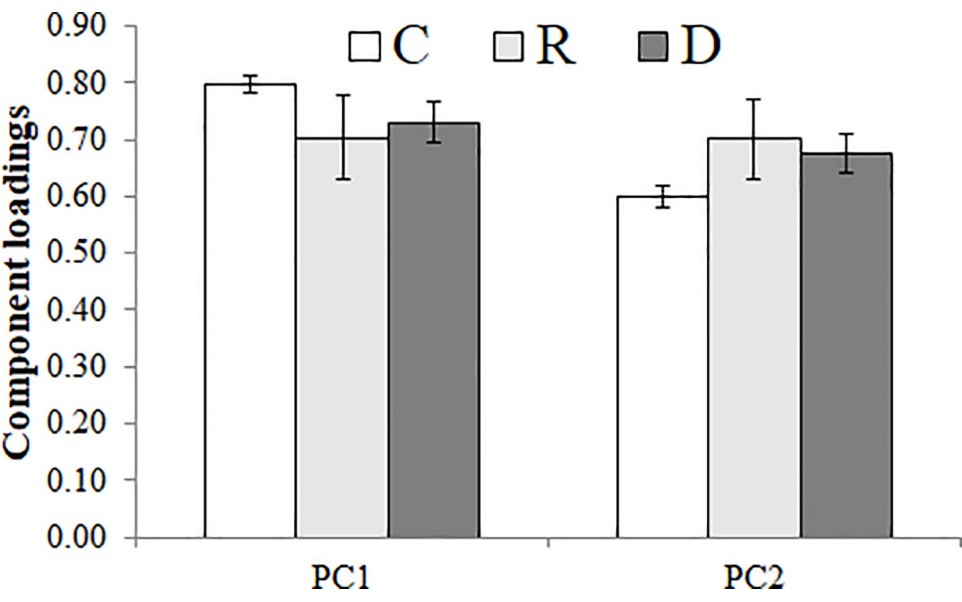

**Fig 5. The bar-plot of PC1 and PC2 loadings of the transposed data matrix PCA.** C for control samples; R for recovered samples; and D for died samples.

**Table 3. One-way ANOVA test between wood test of PC1 and PC2 loadings; and post hoc test of Student–Newman–Keuls between samples (groups are classed in ascending order with the label a, b, c,. ...).**

|  | F value | Pr (>F) | C | R | D |
|---|---|---|---|---|---|
| **PC1** | 9.89 | $2.93 \times 10^{-4}$ | $0.79^b$ | $0.70^a$ | $0.73^a$ |
| **PC2** | 12.73 | $4.54 \times 10^{-5}$ | $0.60^a$ | $0.70^b$ | $0.68^b$ |

In Table 5, the higher standardized discriminant coefficients are indicative of FTIR bands that are most determinants of the discriminant functions, which permitted to differentiate between the three types of samples. DF1 was characterised by bands near 1053, 1193, 1480 and 1521 cm⁻¹ with positive coefficients; and bands near 1103, 1348 and 1542 cm⁻¹ with negative coefficients. And DF2 was characterised by bands near 1035, 1325, 1451, and 1606 cm⁻¹ with positive coefficients; and bands near 1020, 1220 and 1610 cm⁻¹ with negative coefficients. The highest absolute value of the standardised coefficients pointed out DF1 is dominated by the coefficient related to the bands near 1053, 1103, 1348 and 1542 cm⁻¹; whereas it indicated that

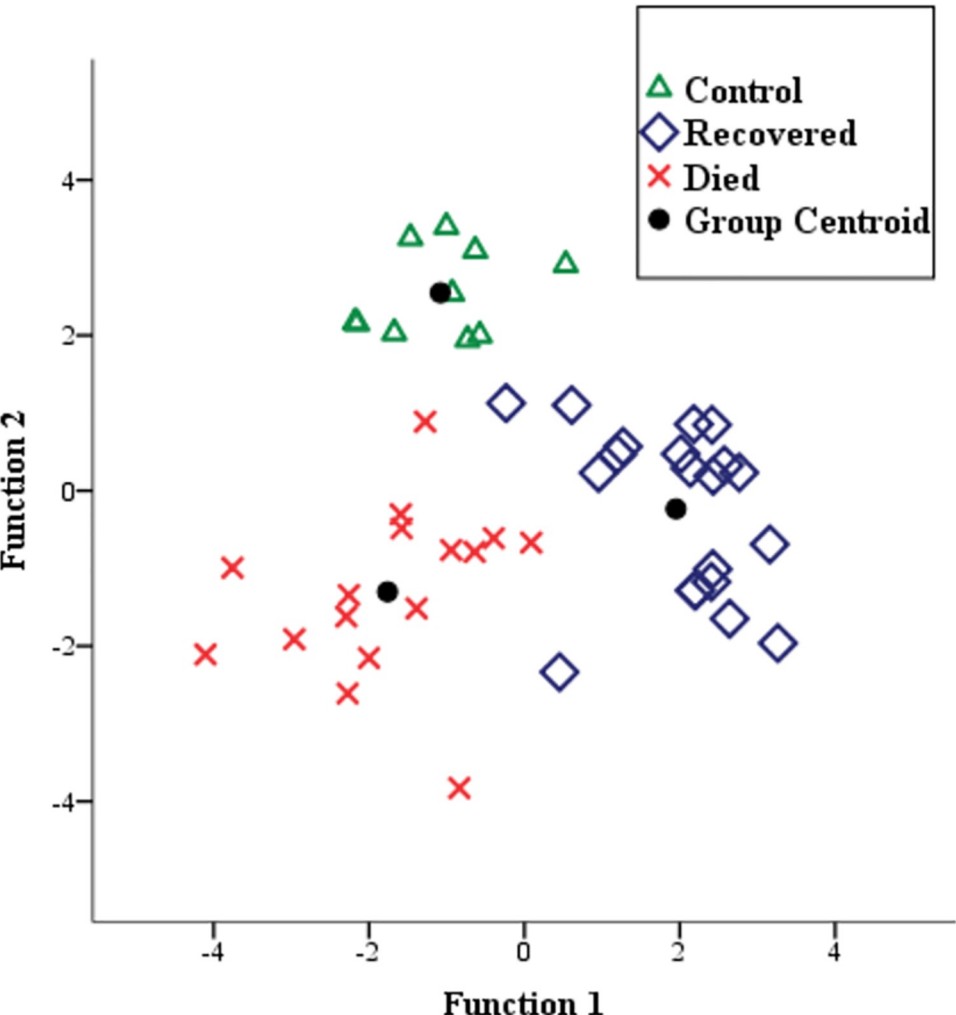

**Fig 6. Plot of the canonical functions obtained by the discriminant analysis with specific FTIR bands of PC1 and PC2.**

**Table 4. One-way ANOVA test between wood test of the two discriminant functions (DF1 and DF2); and post hoc test of Student–Newman–Keuls between samples (groups are classed in ascending order with the label a, b, c).**

|  | F value | Pr (>F) | C | R | D |
|---|---|---|---|---|---|
| **DF1** | 68.84 | $3.94 \times 10^{-14}$ | -1.08[a] | 1.95[b] | -1.76[a] |
| **DF2** | 46.57 | $1.73 \times 10^{-11}$ | 2.55[c] | -0.23[b] | -1.30[a] |

DF2 is dominated by the coefficient related to the bands near 1035, 1020, 1606 and 1610 cm$^{-1}$. Considering these bands with the highest absolute value of the standardized coefficient, the two spectral region mostly susceptible to provide explanation about the differences between these samples are between 1730 and 1580 cm$^{-1}$; and between 1115 and 880 cm$^{-1}$.

## Discussion

The average FTIR spectra at the fingerprint region (1800–800 cm$^{-1}$) show similarity of the molecular contents between the three groups of samples (D, R and C). This resemblance is mainly related to the infrared bands that refer to the main molecular structures of lignocellulosic biomass [33]. Overall, the control wood samples indicated lower intensity for lignin assigned bands, whereas, they indicated higher peaks intensities for polysaccharide assigned bands [33, 34]. In the case of the standard deviation spectra, the spectra on the other hand indicated isolated differences marked by higher peak intensities variation around specific FTIR bands for the recovered branch samples. This variation is probably indicative of a possible modification in the chemical composition of the recovered wood samples. This may suggest the occurrence of biochemical process to resist the influence of woodborer infestation. Previous studies in the literature reported the effect of wood degradation by the variation in peak intensities of bands associated with lignin and polysaccharide compounds [35–37].

The PCA results indicate that the FTIR data related to polysaccharide and lignin contents can be consider to explain the influence of woodborer infestation. For instance, PC1 showed that the control wood samples present higher relative polysaccharide contents. Whereas PC2 showed that the control wood present lower relative lignin contents. This indicates that the

**Table 5. Standardized canonical discriminant function coefficients for DF1 and DF2 (S).**

| wn | DF1 | DF2 |
|---|---|---|
| **1020** | -2.81 | **-11.01** |
| **1035** | 3.18 | **14.76** |
| **1053** | **4.13** | -0.52 |
| **1103** | **-3.34** | -2.97 |
| **1193** | **1.64** | 1.43 |
| **1220** | -1.02 | **-1.71** |
| **1325** | 0.65 | **1.02** |
| **1348** | **-2.61** | -1.43 |
| **1408** | -0.80 | 0.81 |
| **1445** | 0.77 | 0.01 |
| **1451** | 0.33 | **1.43** |
| **1480** | **1.03** | 0.38 |
| **1521** | **1.58** | -0.74 |
| **1542** | **-2.16** | -0.27 |
| **1606** | -1.38 | **2.71** |
| **1610** | 2.37 | **-3.59** |

impact of woodborer moth on this mangrove species results in inducing changes in the relative proportions of its wood molecular contents. In fact, this molecular disturbance putting accent on lignin contents can be interpreted by a possible selective change on the polysaccharide contents. As a matter of fact, polysaccharide compounds are known to be more susceptible to structural transformation than lignin molecules [38]. Also, the observed changes that affect the polysaccharide relative proportion can be a result of a direct influence of wood-boring moth; and also of the biosynthesis of lignin through carbohydrates [39, 40].

Using the infrared bands identified by the PCA factors the discriminant analysis was able to separate the three groups of samples with bands associated to polysaccharide and lignin compounds according to two discriminant factors (DF1 and DF2). For this section, discussion is focused on the FTIR bands that belong to the two spectral regions identified as potential for this study (Fig 3). Results in the present paper suggest that infestation by wood-boring moth could induce more impact on the polysaccharide compounds than the lignin compounds because of its molecular structures [41]. However, it is also possible that during disturbance like the case of insect infestation, biosynthesis of further molecules occurs for a physiological adaptation of the plants [42]. Additionally, polysaccharide compounds constitute larval food resources in many lepidopterans [43, 44]. For instance, research on Lepidoptera gut fluids revealed the occurrence of hydrolytic activities towards cellulose as the main digestion process [45]. Also, Grehan (1988) [46] indicated the potential of larvae to digest $\alpha$-(1, 4) glucans, through a study on gut activity to starch. Important parts of the accessible elements from the nutritional requirement of these insects are contained in the phloem tissues [37].

About 65% of the organic carbon in mangrove wood is offered by carbohydrate compounds. In most of the mangrove species, glucose is presented as the most abundant sugar monomer within carbohydrate compounds [40, 47]. Lepidopteran species were shown as major consumers of mangrove wood polysaccharide compounds in xylem and phloem [48]. In addition to the damage caused on tree stems, it has been shown that wood-boring insects also indirectly cause the mortality of leaves. This phenomenon contributes to increase the infestations and then possibly leading to killing the tree [49, 50].

## Conclusion

Our findings highlighted polysaccharide compounds as the main compounds that undergo changes due to this woodborer infestation. This impact on the polysaccharide compounds has been related to its consumption by this phloem feeding insect. Also our attention was taken to carbohydrates as an important substrate of lignin biosynthesis, during wood formation process. We can therefore conclude that during infestation, *S. alba* reacts by increasing the biosynthesis of Lignin as a way to discourage and stop further infestation. Hence the lignin content in the recovered branches were higher than in the control and the dead.

## Supporting information

**S1 File.**
(XLSX)

## Author Contributions

**Conceptualization:** Nico Koedam.

**Data curation:** Elisha Mrabu Jenoh, Mohamed Traoré.

**Formal analysis:** Elisha Mrabu Jenoh, Mohamed Traoré.

**Funding acquisition:** Elisha Mrabu Jenoh, Nico Koedam.

**Investigation:** Elisha Mrabu Jenoh.

**Methodology:** Elisha Mrabu Jenoh, Mohamed Traoré.

**Project administration:** Elisha Mrabu Jenoh.

**Resources:** Nico Koedam.

**Supervision:** Nico Koedam.

**Validation:** Nico Koedam.

**Writing – original draft:** Elisha Mrabu Jenoh, Mohamed Traoré, Nico Koedam.

**Writing – review & editing:** Mohamed Traoré, Charles Kosore, Nico Koedam.

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
