## [Decision Letter · Decision Letter 0]

19 Aug 2021

PONE-D-21-20278

Biochemical response of Sonneratia alba Sm. branches infested by a wood boring moth (Gazi Bay, Kenya)

PLOS ONE

Dear Dr. Jenoh,

Thank you for submitting your manuscript to PLOS ONE. After careful consideration, we feel that it has merit but does not fully meet PLOS ONE’s publication criteria as it currently stands. Therefore, we invite you to submit a revised version of the manuscript that addresses the points raised during the review process.

We look forward to receiving your revised manuscript.

Kind regards,

Mohammad Shahid, Ph.D.

Academic Editor

PLOS ONE

Journal Requirements:

“This research was funded by the Flemish Interuniversity Council—University Development Cooperation (VLIR-UOS; http://www.vliruos.be/) ICP Ph. D Scholarship 2011. The funder had no role in study design, data collection and analysis, decision to publish, or preparation of the manuscript”

We note that you have provided funding information within the Acknowledgements Section. Please note that funding information should not appear in the Acknowledgments section or other areas of your manuscript. We will only publish funding information present in the Funding Statement section of the online submission form.

“This research was funded by the Flemish Interuniversity Council—University Development Cooperation (VLIR-UOS; http://www.vliruos.be/)ICP Ph. D Scholarship 2011. The funder had no role in study design, data collection and analysis, decision to publish, or preparation of the manuscript.”

5. We note that Figure 1 in your submission contain map images which may be copyrighted. All PLOS content is published under the Creative Commons Attribution License (CC BY 4.0), which means that the manuscript, images, and Supporting Information files will be freely available online, and any third party is permitted to access, download, copy, distribute, and use these materials in any way, even commercially, with proper attribution. For these reasons, we cannot publish previously copyrighted maps or satellite images created using proprietary data, such as Google software (Google Maps, Street View, and Earth). For more information, see our copyright guidelines: http://journals.plos.org/plosone/s/licenses-and-copyright.

6. We note that Figure 2 in your submission contain copyrighted images. All PLOS content is published under the Creative Commons Attribution License (CC BY 4.0), which means that the manuscript, images, and Supporting Information files will be freely available online, and any third party is permitted to access, download, copy, distribute, and use these materials in any way, even commercially, with proper attribution. For more information, see our copyright guidelines: http://journals.plos.org/plosone/s/licenses-and-copyright.

Reviewers' comments:

Reviewer's Responses to Questions

**Comments to the Author**

1. Is the manuscript technically sound, and do the data support the conclusions?

Reviewer #1: Partly

Reviewer #2: Partly

2. Has the statistical analysis been performed appropriately and rigorously? 

Reviewer #1: Yes

Reviewer #2: No

3. Have the authors made all data underlying the findings in their manuscript fully available?

Reviewer #1: Yes

Reviewer #2: Yes

4. Is the manuscript presented in an intelligible fashion and written in standard English?

Reviewer #1: No

Reviewer #2: Yes

5. Review Comments to the Author

Reviewer #1: Sonneratia alba (S. alba) is an important mangrove plant grown in low intertidal zones of downstream estuarine systems of East Africa, Southeast Asia, northern Australia, Borneo and Pacific Islands, etc. The optimal growth of this plant reaches 5 to 50% seawater, indicating its capacity to tolerate high salinity and hypoxia. In this way, S. alba is ecologically significantly important. The infestation by a lepidopteran moth wood borer species is causing mortality of S. alba and create a huge ecological and economic loss. In this aspect, the topic of the present manuscript is very important and is a need of the current situation. Hence, I think the authors bring up some important issues that will spark considerable debate.

However, the manuscript needs some changes and justification, which should be addressed to improve the quality of the paper. I do recommend this manuscript to be published in "Plos One" with major revision and the author/s need to address below comments/suggestions:

1. The work of the present manuscript is focused on the mangrove tree Sonneratia alba, having huge ecological importance. At the beginning of the introduction, the authors mentioned the name of the plant; however, the name of the family is missing. Here, authors need to mention it at an appropriate place because with the plant's name, giving its family name is essential.

2. In the section “Sampling sites and samples,” the authors collected samples from different ages of plants. In certain groups (i.e., A and C), the age of the plants is mentioned, while in the group B and D age of the plants is not given. Here, the authors need to mention the age of all plants.

3. In the material and method section, the author did not mention the time (month and year) of the sampling. The authors need to mention it.

4. It is a well-established fact that in plants, the lignin content varies with the age of the plants. In this manuscript, authors mentioned that lignin content plays a significant role in recovery and damaging from infestation. Hence, authors need to justify the results as they collected samples from different age plants.

5. In Fig. 3, i.e., FTIR spectrum, the authors need to mention the functional groups corresponding to each of the FTIR ranges.

6. The language of the manuscript is poor, and lots of grammatical and typo errors have been seen throughout the manuscript. Authors need to check and reframe the sentences critically.

Reviewer #2: The article entitled "Biochemical response of Sonneratia alba Sm. branches infested by a wood boring

moth (Gazi Bay, Kenya)" should be revised well upon the follwoing:

1- English Editing should be peformed

2- Statistical analysis should be more clear that represents the data collected

3- Refrances should be arranged as PLOS ONE style

6. PLOS authors have the option to publish the peer review history of their article (what does this mean?). If published, this will include your full peer review and any attached files.

Reviewer #1: **Yes: **Mantasha I.

Department of Bio-Molecular Sciences, University of Mississippi, MS, USA

Reviewer #2: No

---

## [Author Response · Author response to Decision Letter 0]

1 Oct 2021

Dear Editor,

I wish to resubmit our manuscript titled biochemical response of Sonneratia alba Sm. branches infested by a wood boring moth (Gazi Bay, Kenya). We have considered all the required changes suggested by the two reviews as much as it is possible. Below is the list the reviewers asked (in blue fonts) and our reaction towards the raised concerns (in Red fonts).

A bout Fig. 1 and Fig. 2, The map layer has been well referenced to Kenya Marine and Fisheries Research Institute (KMFRI). This is a governmental research institute where I am an employee. Being an employee I have full rights to use the map layers as long as I acknowledge it on the map. Which I have done. 

AS for Fig. 2, I wish to say that this is my figure and thus no permission is needed for me to use it.

Reviewer Comments 

Reviewer #1: 

Sonneratia alba (S. alba) is an important mangrove plant grown in low intertidal zones of downstream estuarine systems of East Africa, Southeast Asia, northern Australia, Borneo and Pacific Islands, etc. The optimal growth of this plant reaches 5 to 50% seawater, indicating its capacity to tolerate high salinity and hypoxia. In this way, S. alba is ecologically significantly important. The infestation by a lepidopteran moth wood borer species is causing mortality of S. alba and create a huge ecological and economic loss. In this aspect, the topic of the present manuscript is very important and is a need of the current situation. Hence, I think the authors bring up some important issues that will spark considerable debate.

****However, the manuscript needs some changes and justification, which should be addressed to improve the quality of the paper. I do recommend this manuscript to be published in "Plos One" with major revision and the author/s need to address below comments/suggestions:

1. The work of the present manuscript is focused on the mangrove tree Sonneratia alba, having huge ecological importance. At the beginning of the introduction, the authors mentioned the name of the plant; however, the name of the family is missing. Here, authors need to mention it at an appropriate place because with the plant's name, giving its family name is essential.

Response: This has been done (see line 43)

2. In the section “Sampling sites and samples,” the authors collected samples from different ages of plants. In certain groups (i.e., A and C), the age of the plants is mentioned, while in the group B and D age of the plants is not given. Here, the authors need to mention the age of all plants. 

Response: It is true that the age of the trees at some of the plots have been given in the materials and methods section whereas other plots (B and D) have not been given. The reason for this is actually given in the materials and method section where the description of the sampling plots was given. Plot B and D are natural forest that have been since time immemorial whereas plot A and C are plantation. Thus information about these two plots is available. On the other hand, in this research, as indicated in the manuscript, all efforts were done to ensure the samples were taken from the same canopy height and the branches were of similar size as much as possible. 

Also it is difficult to give the general age of mangrove trees using readily available tools in plant science. This is because methods of age estimation in mangrove have not been fully developed as is the case in other trees in terrestrial forest. It is our belief that even providing the DBH does not add any scientific relevance since a fully grown mangrove may stagnant at some stage. 

3. In the material and method section, the author did not mention the time (month and year) of the sampling. The authors need to mention it. 

Response: This has been done as requested (see lines 115-116)

4. It is a well-established fact that in plants, the lignin content varies with the age of the plants. In this manuscript, authors mentioned that lignin content plays a significant role in recovery and damaging from infestation. Hence, authors need to justify the results as they collected samples from different age plants.

Response: As much as the samples were collected from trees of various ages, the number of cumulative samples was made large enough and equal between the three categories i.e. (Dead, recovered and control). To ensure lack of bias, it has been mentioned in the manuscript that all effort was put to ensure that sampled branches were taken from same canopy position and of similar size in every sampled tree. Thus the sampled branches were assumed to be of similar physiological and developmental processes in each sampled tree. The concluded results were therefore out of the influence of age and size due to the replication effect. 

5. In Fig. 3, i.e., FTIR spectrum, the authors need to mention the functional groups corresponding to each of the FTIR ranges.

Response: This suggestion has been considered. See new version of Fig 3.

6. The language of the manuscript is poor, and lots of grammatical and typo errors have been seen throughout the manuscript. Authors need to check and reframe the sentences critically.

Response: We have thoroughly revised the manuscript and made many corrections, as can be observed in the Changes marked version of the revised manuscript.

Reviewer #2: 

The article entitled "Biochemical response of Sonneratia alba Sm. branches infested by a wood boring moth (Gazi Bay, Kenya)" should be revised well upon the follwoing:

1- English Editing should be performed

Response: We have thoroughly revised the manuscript and made many corrections, as can be observed in the Changes marked version of the revised manuscript.

2- Statistical analysis should be more clear that represents the data collected

Response: this comment from the reviewer seems not clear from our opinion. However, we would like to mention that the statistical methods applied in this paper (PCA and LDA) are very commonly used methods for such research work. Furthermore, the coherence in the results is reflected by the fact that the results of PCA and LDA support each other. In addition, we applied the One-way ANOVA test to assess the statistical significance of the results.

3- Refrances should be arranged as PLOS ONE style

Response: This suggestion has also been considered according to the journal requirements. 

We re-submit this work after carefully considering all the comments of the reviewers and having done the necessary changes. We once again submit that this work has not been prior submitted to any other journal for publication consideration. 

On behalf of all authors,

*Elisha Mrabu Jenoh

Kenya Marine and Fisheries Research Institute

P. O. Box 81651 (80100)

Mombasa Kenya

E-mail: elishamrabu@gmail.com or emrabu@kmfri.co.ke

---

## [Decision Letter · Decision Letter 1]

18 Oct 2021

Biochemical response of Sonneratia alba Sm. branches infested by a wood boring moth (Gazi Bay, Kenya)

PONE-D-21-20278R1

Dear Dr. Jenoh,

We’re pleased to inform you that your manuscript has been judged scientifically suitable for publication and will be formally accepted for publication once it meets all outstanding technical requirements.

Kind regards,

Mohammad Shahid, Ph.D.

Academic Editor

PLOS ONE

Additional Editor Comments (optional):

Reviewers' comments:

Reviewer's Responses to Questions

**Comments to the Author**

1. If the authors have adequately addressed your comments raised in a previous round of review and you feel that this manuscript is now acceptable for publication, you may indicate that here to bypass the “Comments to the Author” section, enter your conflict of interest statement in the “Confidential to Editor” section, and submit your "Accept" recommendation.

Reviewer #1: All comments have been addressed

Reviewer #2: All comments have been addressed

2. Is the manuscript technically sound, and do the data support the conclusions?

Reviewer #1: Yes

Reviewer #2: Partly

3. Has the statistical analysis been performed appropriately and rigorously? 

Reviewer #1: Yes

Reviewer #2: Yes

4. Have the authors made all data underlying the findings in their manuscript fully available?

Reviewer #1: Yes

Reviewer #2: Yes

5. Is the manuscript presented in an intelligible fashion and written in standard English?

Reviewer #1: Yes

Reviewer #2: Yes

6. Review Comments to the Author

Reviewer #1: Journal- PLOS ONE

Manuscript ID- PONE-D-21-20278

Title- Biochemical response of Sonneratia alba Sm. branches infested by a wood boring

moth (Gazi Bay, Kenya)

Dear Editor,

Authors have fulfilled all the queries/comments as it was asked by reviewers previously. Hence, now the manuscript is well written. I believe that it is a nice piece of work for being published in the PLOS ONE. Finally, I recommend that the paper should be accepted for the publication in the present form.

Decision- Accept

Reviewer #2: (No Response)

7. PLOS authors have the option to publish the peer review history of their article (what does this mean?). If published, this will include your full peer review and any attached files.

Reviewer #1: No

Reviewer #2: No

---

## [Editor Report · Acceptance letter]

22 Oct 2021

PONE-D-21-20278R1 

Biochemical response of *Sonneratia alba* Sm. branches infested by a wood boring moth (Gazi Bay, Kenya) 

Dear Dr. Jenoh:

I'm pleased to inform you that your manuscript has been deemed suitable for publication in PLOS ONE. Congratulations! Your manuscript is now with our production department. 

Kind regards, 

on behalf of

Dr. Mohammad Shahid 

Academic Editor

PLOS ONE